# High-Performance Flexible Piezoresistive Pressure Sensor Printed with 3D Microstructures

**DOI:** 10.3390/nano12193417

**Published:** 2022-09-29

**Authors:** Guohong Hu, Fengli Huang, Chengli Tang, Jinmei Gu, Zhiheng Yu, Yun Zhao

**Affiliations:** 1College of Mechanical Engineering, Zhejiang University of Technology, Hangzhou 310014, China; 2College of Information Science and Engineering, Key Laboratory of Advanced Manufacturing Technology of Jiaxing City, Jiaxing University, Jiaxing 341000, China; 3College of Mechanical and Electrical Engineering, Jiaxing Nanhu University, Jiaxing 314000, China

**Keywords:** flexible pressure sensors, intermediate dielectric layer, microstructure array, aerosol printing, sensing performance

## Abstract

Flexible pressure sensors have been widely used in health detection, robot sensing, and shape recognition. The micro-engineered design of the intermediate dielectric layer (IDL) has proven to be an effective way to optimize the performance of flexible pressure sensors. Nevertheless, the performance development of flexible pressure sensors is limited due to cost and process difficulty, prepared by inverted mold lithography. In this work, microstructured arrays printed by aerosol printing act as the IDL of the sensor. It is a facile way to prepare flexible pressure sensors with high performance, simplified processes, and reduced cost. Simultaneously, the effects of microstructure size, PDMS/MWCNTs film, microstructure height, and distance between the microstructures on the sensitivity and response time of the sensor are studied. When the microstructure size, height, and distance are 250 µm, 50 µm, and 400 µm, respectively, the sensor shows a sensitivity of 0.172 kPa^−1^ with a response time of 98.2 ms and a relaxation time of 111.4 ms. Studies have proven that the microstructured dielectric layer printed by aerosol printing could replace the inverted mold technology. Additionally, applications of the designed sensor are tested, such as the finger pressing test, elbow bending test, and human squatting test, which show good performance.

## 1. Introduction

The emerging industry of flexible electronics has greatly promoted the development of medical monitoring [1,2,3], artificial intelligence [4], life and health [5,6,7], and the application of robots [8]. Moreover, flexible electronics will also have great application advantages in public security, aerospace, national defense, and military industry [9,10,11,12]. At the same time, flexible electronics will go further in the direction of integration [13,14], wearables [15,16], and light weight.

With the advancement of industrial technology and the development of modern science and technology, more demands have been put forward for the performance of sensors. Simply depositing conductive nanomaterials [17,18,19,20,21,22] and carbon-based materials [23,24,25,26,27] on flexible substrates or mixing elastic substrates with conductive fillers to form new dense active layers cannot further improve sensing performances, such as sensitivity and sensing range. Therefore, higher requirements for the structure of the intermediate active layer have been proposed so the structure of the intermediate active layer could be designed to meet the demands of further improvement of the sensing performance. The main micro-nano-design structures of the intermediate active layer are summarized as follows: microwaves [28], microprotrusions [29], micropyramids [30], micropillars [31], biomimetic microstructures [32,33,34], porous foam microstructures [35], and composite microstructures where multiple microstructures coexist [36]. Each type of microstructural engineering provides different advantages for specific applications. At the same time, the versatility and tunability of the size and shape of the microstructures are very valuable when trying to tune the sensitivity of the sensor [37]. A large number of experiments have shown that the micro-nano-structure could significantly improve the sensitivity, detection limit, response time, and other performances of the flexible sensor. Thus, many teams have microengineered the active layers of capacitive [38,39], resistive [40,41], piezoresistive [42,43], piezoelectric [44], and tribological pressure sensors [45] to further improve sensor performance. In 2016, Li et al. [46] used the biomimetic microstructure of the natural lotus leaf to design and manufacture a new high-performance flexible capacitive tactile sensor. Using the unique lotus leaf surface micropattern as the electrode template and polystyrene microspheres as the dielectric layer, the designed device had high sensitivity (0.815 kPa^−1^), wide dynamic response range (0~50 N), stable high sensing performance, and fast response time (38 ms). Furthermore, flexible capacitive sensors are suitable for not only pressure (touching a hair) but also bending and stretching forces. In 2017, Cheng et al. [47] fabricated an inverted pyramid structure mold with photolithography and anisotropic wet etching, and proposed novel pressure sensor design methods, using a layered micropyramid structure to significantly reduce hysteresis, maintaining its high sensitivity. In 2018, Pang et al. [48] obtained randomly distributed spine-like microstructures through the combination of sandpaper templates and reduced graphene oxide (r-GO). The randomly distributed spine-like-structure (RDS) graphene pressure sensor had a sensitivity of up to 25.1 kPa^−1^ over a wide linear range of 0 to 2.6 kPa^−1^. Simulation and mechanistic analysis showed that the spine-like structure and random distribution contributed to the high sensitivity and large linear range, respectively. In 2021, Luo et al. [49] fabricated PVDF/SWCNTs thin films for piezoelectric pressure sensors by near-field electro-hydraulic direct writing. By adopting the technological method of moving the rectangular track, the micro-nano-optical fiber array was directly programmed to make a piezoelectric pressure active film, so the piezoelectric film had structural consistency and improved the piezoelectric performance. The designed flexible piezoelectric pressure sensor had a high sensitivity (15.68 kPa^−1^) and fast response time (66 ms). In general, all elastic microstructured films with opposite characteristics to the mold microstructure were based on artificial silicon wafer molds or natural biomaterials, prepared by conventional templating techniques. The high-sensitivity flexible sensor prepared by the natural biological template solved the problem of a single structure of the intermediate active layer; however, the shape and size of the microstructure prepared in this way were all affected by the random distribution of the internal structure of the organism, which had irregularity, inhomogeneity, and irreversibility. Moreover, one organism could only replicate one microstructure, which greatly limited the diversification of the microstructure design. Meanwhile, the microstructure mold prepared by lithography design had a complicated process and was highly dependent on the equipment, which led to a high-cost manufacturing process; the shape and size of the lithography mold were also determined, resulting in a microstructure that was inverted. The size of the shape was simplistic. Of course, the microstructure cast by sandpaper could effectively avoid the problem of high manufacturing cost, but the nonuniformity of the structure form would lead to uncontrollable sensing performance of the prepared sensor. It has been proven that the more the shape, size, and spacing of the microstructures are controlled, the more the performance of the sensor is controlled, which will be of great significance to the development of the sensor. Unlike the traditional mold process, the intermediate layer structure printed by aerosol printing technology had proven that the intermediate microstructure layer could be 3D-printed. However, the currently studied aerosol printing technology could only print a dense thin film layer, which was far from meeting the requirements of sensor performance improvement and controllability as the dielectric layer of the sensor.

Based on aerosol printing technology and the sandpaper inverted mold method, in this work, a new method for preparing a high-sensitivity flexible piezoresistive pressure sensors was demonstrated at low cost. In the experiment, a silver interdigital electrode (IDE) was printed on a PET flexible substrate by Electro-Hydro Dynamics (EHD) near-field direct writing technology (laboratory-independent research and development equipment). Then, the square microstructure was printed in the middle of the silver IDE using aerosol printing technology. Finally, PDMS/MWCNTs film was packaged with sandpaper inverted molds, obtaining a flexible pressure sensor with a bilayer structure. At the same time, the experiment explored the effect of the distance between microstructures on the response time and sensitivity of the sensor, and the influence of the size and height of the microstructure on the sensitivity of the sensor. In order to further explore the application of the sensor in human motion recognition, corresponding experiments were also carried out.

## 2. Materials and Methods

### 2.1. Materials

Poly (3,4-ethylenedioxythiophene): Poly (styrene sulfonate) (Shanghai Macklin Biochemical Co., Ltd., Shanghai, China) was used to prepare an organic conductive polymer mixture as the printing ink for experiments. Triton X-100 (Sinopharm Chemical Reagent Co., Ltd., Shanghai, China) and FS-30 (Shenzhen Laikeer Chemical New Materials Co., Ltd., Shenzhen, China) were used as surfactants. Triton X-100 and FS-30 with a volume fraction of 0.1% and 0.2%, respectively, were added into the conductive polymer mixed solution to reduce the surface tension of the solution, and the surface tension of the obtained ink reached the range of 28 mN/m to 42.3 mN/m, which was conducive to the formation of stable-diameter printing lines on PET flexible substrates. The above mixed solution was sealed and placed in an ultrasonic dispersion cell for 2 h, and the mixed solution was used for printing experiments after homogeneous dispersion. Using PET film as a flexible insulation substrate, the surface tension range of the printing ink acceptable for printing was between 20 mN/m and 40 mN/m. PDMS/MWCNTS films were prepared by the PDMS main agent (10:1 ratio of main agent to curing agent, Trademark of The Dow Chemical Company, Michigan, MI, USA), PDMS curing agent (Sylgard 184, Trademark of The Dow Chemical Company, Michigan, MI, USA), and MWCNTs (diameter 110–190 nm, length 5–9 μm, Sigma Aldrich (Shanghai) Trading Co., Ltd., Shanghai, China). In addition, 50 wt% silver nanoparticle printing ink (Sigma Aldrich (Shanghai) Trading Co., Ltd., Shanghai, China) was used to print the silver IDE. Isopropanol (IPA) (Shanghai Macklin Biochemical Co., Ltd., Shanghai, China) served as the dispersant for MWCNTs. Conductive copper foil (3J218, Dongguan Xinshi Packaging Materials Co., Ltd., Dongguan, China) was used as output wire.

A high-precision aerosol printing system and Nano-Jet (Yi Xin Technology Co., Ltd., Shenzhen, China), and EHD near-field direct-write printing equipment, independently developed by the laboratory, were used as the printing equipment.

### 2.2. Methods

#### 2.2.1. Printing of Silver IDEs and Microstructure Arrays

An EHD near-field direct-writing printing device was used to print 50 wt% silver nanoparticle conductive ink on a flexible-substrate PET film to prepare a silver IDE.

The line width and uniformity of the printed silver IDE obtained a higher resolution, which was controlled through the adjustment of printing voltage, printing speed, ink flow, and other parameters. The distance between the silver IDE was adjusted by G-code numerical control programming, so the microstructure array could be printed in the middle of the line between the spacing of electrodes. Then, the PET film with the printed silver IDE was put into the vacuum drying box and dried at room temperature for 6 h until it was dried completely. Then, the dried silver IDE was removed and put into the hot-pressing equipment, which was coated with a PI film up and down to prevent the silver IDE from directly touching the hot-pressing substrate to be damaged. Meanwhile, the hot-pressing temperature was controlled at 120 °C, the pressure was 9000 N, and the time was 10 min. Finally, a silver IDE layer, with low resistivity and suitable for the next experiment, was obtained.

As shown in Figure 1a,b, the microstructure printing schematic and the physical diagram of the aerosol equipment are presented. The sintered PET film was fixed on the mobile substrate of the aerosol printing equipment, and the multi-layer printing was carried out with the prepared PEDOT:PSS mixed solution. The printed one-dimensional (1D) lines were filled with 2D graphic structures by G-code programming. At the same time, due to the use of Aerosol Near-Field Printing Technology, the direct writing printing height was set to 1 mm, and the inner diameter of the print needle was 210 µm, through which the high-precision positioning of the printed pattern could be achieved. Therefore, after the 2D graphics were printed, the multi-layer stacking printing of the 2D graphic structures into 3D microstructures with a certain height after repeated positioning was accomplished, completing the transformation from straight lines to graphics to 3D microstructures. In the experiment, a square with a microstructure diameter of 400 µm and a height of 20 µm was printed. The individual square microstructures were then programmed by the G-code program, forming a well-arranged array of 1 cm × 1 cm microstructures and evenly arranged in the spacing of the printed silver IDE. The distance between the center of the square and the center of the other one was controllable. Based on the G-code program, the distance to change the density of the square microstructure could be controlled easily.

#### 2.2.2. Preparation of PDMS/MWCNTs Thin Films

The elastoplasticity of the material was determined by the size of the Young’s modulus, so the smaller the Young’s modulus, the better the elasticity. However, the Young’s modulus of PDMS was 2 MPa, and it had good elasticity, so it could quickly produce deformation when under force, and could quickly return to its original state when unloading. This characteristic could contribute to improve the response time of the sensor.

The preparation process of PDMS/MWCNTs film is shown in Figure 1c. First, 8 wt% MWCNTs was added to isopropanol, sealed, and placed on a magnetic stirrer for 2 h at room temperature, so MWCNTs could be evenly dispersed into isopropanol. After the dispersion completed, it was removed and 4 g of PDMS main agent was added. Then, the mixed solution was first magnetically stirred at room temperature for 30 min to make it fully mixed; then, the oil bath temperature was adjusted to 60 °C, and the magnetic stirring was continued for 2 h so that the isopropanol solution was completely volatilized. This was removed after magnetic stirring, and 0.4 g of PDMS curing agent was added into the solution and stirred well with a glass rod. The appropriate size of sandpaper was cut and fixed flat on the glass plate, the glass plate was placed on the spin coater platform, and the vacuum extraction pump was opened to fix it. The prepared PDMS/MWCNTs mixed solution was initially and evenly applied on the pre-prepared sandpaper, and the spin coater was set to a speed of 1000 r/min for 2 min. Finally, the PDMS/MWCNTS film with a uniform texture with a thickness of 200 µm was obtained.

#### 2.2.3. Fabrication of Flexible Piezoresistive Pressure Sensor

As shown in Figure 2b, the flexible piezoresistive pressure sensor studied in this experiment consisted of two parts: a flexible substrate PET film with the silver IDE and the microstructure array printed, and the top flexible contact layer of the PDMS/MWCNTs film. The specific preparation process is shown in Figure 2a; first, the IDE was printed on the PET flexible substrate, and then the square microstructure array was printed in the middle of the IDE. A 2 cm × 3 cm PET substrate and a 2 cm × 3 cm PDMS/MWCNTs film were cut, and the upper and lower layers were aligned. Because PDMS had good adhesion, the substrate and the top flexible contact layer could be temporarily fixed, and then the adhesive was used to hold it in place. A suitable size of copper foil was cut and silver paste was used to fix it on the two pins of the silver IDE, which was the output position of the electrical signal. As shown in Figure 2c, the size of the prepared sensor sample was 2 cm × 3 cm. However, the effective working area of the sensor, which was 1 cm × 1 cm, was mainly covered by the microstructures.

### 2.3. Characterization and Measurement

PDMS/MWCNTs films were characterized by field emission scanning electron microscopy (SEM, FEI Inspect F50, Massachusetts, MA, USA). The microstructure was characterized by inverted metallographic microscopy (DMI3000M/DFC450, Leica, Wetzlar, Germany). The height of the square was tested by the Dektak XT step meter (Dektak-XT-10th, Bruker Technology Co., Ltd. Massachusetts, MA, USA). The flexible piezoresistive pressure sensor was electrically characterized by a digital source meter (Keithley 6510, State of Oregon, OR, USA) and an electronic universal material testing machine. The PDMS/SWCNTS film square resistance was determined by using a variable-temperature four-probe test bench (SM-4, Shenzhen Xinzhishang Electronics Co., Ltd., Shenzhen, China).

As shown in Figure 3a–c, the surface structure of the printed-out microstructure could be seen through the microscope, and the size and shape were also in line with the pre-designed G-code shape. As shown in Figure 3d, MWCNTs were evenly dispersed on the surface of the film. As shown in Figure 3e, under a high-power electron microscope, MWCNTs were cross-distributed on the surface of the film, thus forming more conductive paths. As shown in Figure 3f, in the cross-sectional view of the film, MWCNTs were crisscrossed inside the film to form numerous conductive paths, and after compression, the upper and lower MWCNTs were forced to be more closely intertwined and stacked together, thereby enhancing the conductive pathway. Figure 3h shows the height curve of the microstructure step instrument, the height curve of the microstructure was relatively flat, and there were several raised peaks for the G-code program to design the trajectory of the print needle. The overall curve was in a square shape, with its length being the transverse dimensions of the microstructure and the height being the longitudinal dimensions of the microstructure. This further illustrated the complete structure of the printed microstructure and the uniform printing.

## 3. Results and Discussion

### 3.1. Effect of Microstructure Size and PDMS/MWCNTs Film on Sensor Sensitivity

Sensitivity is one of the key technical indicators of the sensor. For a piezoresistive flexible pressure sensor, the sensitivity calculation formula is as follows:*S* = [(*R* − *R*_0_)/*R*_0_]/*ΔP* = (*ΔR*/*R*_0_)/*ΔP*(1)
where *S* is the sensitivity, *∆**R* is the change value of the resistance, *∆P* is the pressure per unit area of the sensor, and the pressure is applied by the Keithley6510 and electronic universal material testing machine, shown in Figure 3h; *R* is the resistance value of the sensor when it is pressurized, and *R*_0_ is the initial resistance value of the sensor in the unpressurized state.

As shown in Figure 4a, there were three main resistances of the sensor, which were the silver IDE resistance *R_p_*, the microstructure resistance *R_c_*, and the PDMS/MWCNTs thin-film resistance *R_d_*. When the pressure *P* = 0, there were fewer passages between the MWCNTs mixed in the PDMS, resulting in a large resistance value of *R_d_*; when the pressure *P* ≠ 0, the PDMS was squeezed, causing more mutual contact between the MWCNTs inside it. MWCNTs formed more conductive paths inside the PDMS, which reduced *R_d_*. Secondly, when the pressure *P* was applied, the total resistance was as shown in the Figure 4a. During the loading process, the number and contact area of the contact with the conical microstructures increased, attributed to the deformation of the PDMS/MWCNTs film layer under the force, and the value of *R_c_* decreased. In the process of unloading, the deformation of the PDMS/MWCNTs film gradually decreased. On the one hand, it caused the reduction in the MWCNTs conduction path and led to the increase in *R_d_*; on the other hand, it led to the decrease in the contact area and the number of contacts between the film and the cone. The decrease caused an increase in the resistance value *R_c_*. Therefore, it could be seen that *∆R* in the above formula was mainly the change in *R_c_* and *R_d_*.

To study the effect of microstructure size on sensor sensitivity, we controlled the height of the microstructures (mainly determined by the number of printed layers) and the distance between the squares (the distance from the edge of the square to the edge), as shown in Figure 4b; square arrays with diameters of 200 µm, 250 µm, 400 µm, and 600 µm were printed. In the experiment, under the action of the same force *P*, the resistance value *∆R* was measured in the process of loading and unloading of the microstructure arrays of various sizes as the IDL sensor was tested by the instrument, respectively. As shown in Figure 4c, under the action of the applied pressure *P*, the sensitivity of M1–M4 gradually decreased with the increase in the square size. It could be seen from the analysis that due to the different sizes of the microstructures, the coverage of the microstructures would increase with the increase in the diameter of the microstructures within the same effective area. On the contrary, the number of squares per unit area would also decrease with the increase in the size of the square, resulting in the reduction in the overall device resistance *R* change, due to the reduction in the conductive path. Thus, the sensitivity decreased. At the same time, it would also cause the reduction in the air porosity, affecting the change in sensitivity to a certain extent. In addition, it is the air porosity or air gap rate where the air volume between the PDMS/MWCNTs thin film and the microstructure is divided by the sum of the air volume and the microstructure volume. The larger the air porosity of the device is, the easier the PDMS/MWCNTs film of the device is to be compressed and deformed when loaded, and the more obvious the resistance change rate caused by deformation is, demonstrating that the sensitivity of the sensor improved.

### 3.2. Influence of Microstructure Height on Sensor Sensitivity

The square microstructure was used as the IDL of the sensor, and the height of the microstructure had a crucial impact on the sensitivity of the sensor. According to the analysis above, the sensitivity of the sensor was determined by *∆R*, while *∆R* was determined by *R_c_*. It could be seen that the size of *R_c_* was mainly determined by the number of contacts between the microstructure and the PDMS/MWCNTs film, as well as the contact area. As shown in Figure 4b, a microstructure array with a height of 16 µm, 32 µm, 50 µm, and 68 µm was printed, subject to the same force *P*, and the change value of its resistance value was measured in the process of force addition and unloading to investigate the effect of the microstructure height on the sensor sensitivity. As shown in Figure 4d, the sensitivity of M5-M7 increased as the height of the square rose from 16 µm to 50 µm; additionally, when the height rose to 68 µm, the sensitivity of M8 showed a downward trend. In short, the mass fraction of PEDOT:PSS in the printing ink was 1.5%, so the conductivity of the ink was low. Meanwhile, with the increase in the height, the resistance *R_c_* of the microstructure increased. However, when the printing height reached 68 µm, the resistance *R_c_* of the microstructure was too large, resulting in a decrease in the overall resistance change rate of the device. Furthermore, the dried microstructure in the height of 68 µm was prone to cracks, leading to an increase in the resistance of the microstructure. These caused a decrease in sensitivity. It could be seen from the analysis that in the initial state, only part of the top surface of the microstructure was in contact with the top film (as shown in Figure 4a). As the pressure load increased, more microstructures came into contact with PDMS/MWCNTs films, and the displacement increased further; the local stress at the top of the microstructure was concentrated, squeezing the sides of the microstructure. With the contact area increasing, the resistance *R_c_* decreased, causing changes in sensitivity. In other words, the higher the height of the microstructure, the higher the number of contacts between the two, but as the height of the microstructure rose, the resistance value of the individual microstructure would increase accordingly, causing the relative increase in the overall resistance *R_c_* to cause a decrease in sensitivity. At the same time, the overall height of the microstructure rose, and the PDMS/MWCNTS film was embedded in the middle of the microstructure, resulting in an increase in the air gap rate, which also had a certain impact on the sensor sensitivity. 

### 3.3. Influence of Distance between Microstructures on Sensor Sensitivity and Response Time

To study the effect of distance between microstructures on sensitivity and response time, the coverage of microstructure area would become a particularly critical issue, affecting the entire experiment. As shown in Figure 4b, a microstructure array with a spacing of 200 µm, 400 µm, 600 µm, 800 µm was printed. Pressure *P* was applied in the experiment, as shown in Figure 4e, and the sensitivity of M9–M12 decreased as the spacing (the distance from the edge of the square to the edge) decreased. As shown in Figure 5a, the spacing increased from 200 µm to 800 µm, the corresponding time of M9-M12 increased from 98.2 ms to 235.3 ms, and the response time increased the most from 200 µm to 400 µm; as the spacing continued to increase, the growth trend of the response time was slowing down. Through the analysis, it could be seen that as the square spacing increased, the number of squares per unit area decreased, which was the main factor in the response time and sensitivity of the device. On the one hand, the decrease in the number of PDMS/MWCNTS thin film layers in contact with the square microstructure and the decrease in contact area led to an increase in the resistance value *R_c_*; on the other hand, this would lead to an increase in the air gap rate and a decrease in the force point of the PDMS/MWCNTS film, resulting in an increase in the stretch deformation of the film and a relative increase in the resistance value *R_d_*, and it was this reason that led to an increase in the response time of the device. In addition, as shown in Figure 4f–h, in the cyclic force test of the device M6 at 5 kPa, the output signal of the device was stable, and an excellent response time (98.2 ms) and recovery time (111.4 ms) were demonstrated; meanwhile, the recovery time was 13.2 ms later than the response time. When working during the cycle of a single force, the deformation of the device was performed, responding immediately. Meanwhile, a certain adhesion phenomenon existed between the PDMS/MWCNTs film and the printed microstructured active layer, resulting in a delay in recovery time during the deformation recovery process of the device.

### 3.4. Relaxation Time and Application of the Device in Human Motion Detection

Another important criterion for a high-performance pressure sensor was a low relaxation time. The relaxation time was the time required for the resistance signal to change from the stable output value to the initial value after the applied pressure was removed. Unlike printing elastically deformable materials with good density as the IDL, the performance of the sensor could be further improved by introducing air gaps as the IDL by printing cylindrical microstructures. Compared with the IDL structure of the dense layer structure, the introduction of air gaps in the IDL of the printed microstructure was conducive to the rapid formation and recovery of deformation. Due to the low Young’s modulus of PDMS material showing good elastic deformation, the top contact layer made of the thin film, which was prepared from the PDMS/MWCNTs mixed solution and combined with the microstructured IDL introduced with the air gap, enabled the sensor, such that the corresponding relaxation time of sensor loading and unloading was within the time scale of minutes and seconds.

In order to further explore the application of the prepared flexible piezoresistive pressure sensor in human motion recognition, corresponding experiments were carried out. As shown in Figure 5b, the device showed a relatively stable cyclic signal output through intermittent finger compression, and the resistance value also showed a steady state when the finger pressed and released for a certain period of time. As shown in Figure 5c, through the control of the finger force, the device was constantly cycling through bending cycles. On the one hand, when the device was bent, the MWCNTs in the PDMS/MWCNTs film were squeezed to contact to form a path so that the resistance was reduced. On the other hand, PDMS/MWCNTs films contacted with the microstructures to form a path that reduced resistance. When the device was restored, the resistance of the device changed in the opposite way, and the resistance increased, and in the cycle of continuous bending in recovery, the device exhibited a stable resistance signal output state. As shown in Figure 5d, the device was tightly pasted to the elbow socket, and the arm was constantly bent and extended. When the elbow was bent, the device was squeezed, the PDMS/MWCNTs film produced compression deformation MWCNTs and formed more conductive paths, and the number of microstructures and the film contact increased, resulting in a sharp change in resistance (*∆R*). As shown in Figure 5e, the device was tightly pasted at the knee of the human body through about 1 s of squatting action, which was the human body from standing to squatting in the process, and the state of the device was from the initial flattening to the compression state after the final squat. During this process, the PDMS/MWCNTs film of the device first produced a distance, increasing in distance of the tensile MWCNTs, and resulting in a decrease in the conductive path and an increase in resistance; with the deepening of the squat process, the PDMS/MWCNTs film was compressed, the number of contacts with the microstructure and the contact area were rapidly increased, and the overall resistance decreased. It was shown that the device had a good output signal during the cyclic movement of the human body squatting. Meanwhile, when testing different joint parts of the human body, the recovery resistance of the device will have a large difference. Because the PDMS/MWCNTs film will stretch during the recovery process after the device is bent, it will result in a decrease in the contact between MWCNTs and cause the original resistance to increase, and the greater the degree of bending, the greater the original resistance.

## 4. Conclusions

In this work, a flexible piezoresistive pressure sensor using the aerosol printing technique and sandpaper casting technique was fabricated. The sensor was composed of a top contact layer and an IDL with an electrode as the bottom layer. The top contact layer was obtained by pouring the mixed solution of PDMS/MWCNTs on sandpaper, and the bottom layer of the sensor was obtained by printing a silver IDE on the flexible substrate PET film by EHD near-field direct writing printing equipment and printing square microstructure arrays by aerosol printing technology. Compared with the traditional “sandwich” structure, the sensor with the bilayer structure simplified the fabrication process, which provides a new way for the design of the sensor structure. It is a facile way to prepare flexible pressure sensors with simplified processes, reduced cost, and high sensing performance. By controlling the height, size, and spacing of the printed microstructures, the sensitivity, response time, and relaxation time of the sensor could be controlled effectively. Additionally, the sensor prepared with a microstructure size of 250 µm, height of 50 µm, and distance of 400 µm had the best performance. The sensitivity of the sensor was 0.172 kPa^−1^, the response time was 98.2 ms, and the relaxation time was 111.4 ms.

Compared with the flexible sensor that was printed a monolithic dense pattern as the IDL by the same process, this work focused on the improvement of the microstructure preparation process. Aerosol printing technology was used to print high-precision microstructures and air gaps were introduced to improve the sensitivity of the sensor and reduce the response time. At the same time, though the top flexible contact layer was prepared with traditional sandpaper casting technology, the combination of PDMS with low Young’s modulus and MWCNTs with high conductivity not only further optimized the performance of the sensor in these two aspects, but also ensured the flexible characteristics of the sensor.

## Figures and Tables

**Figure 1 nanomaterials-12-03417-f001:**
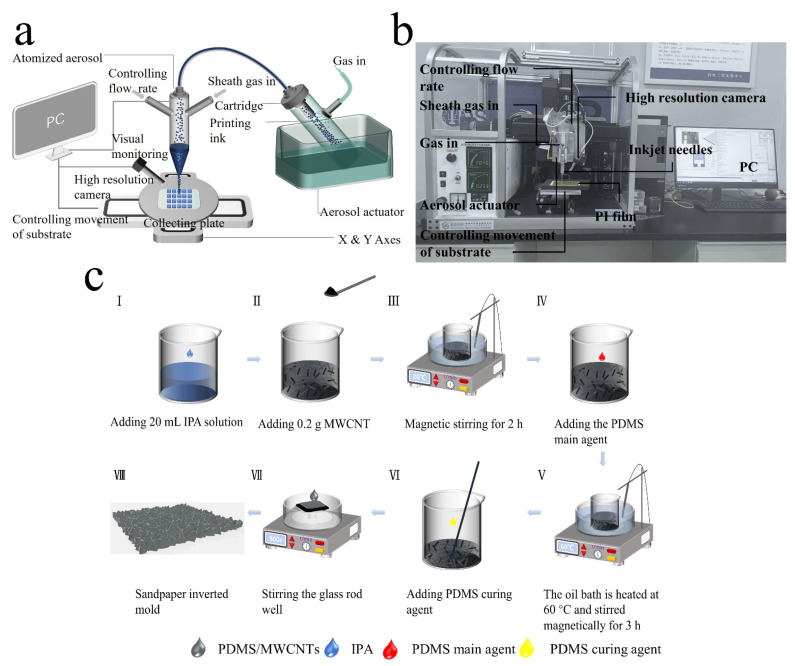
(**a**) Printing principle of aerosol technology. (**b**) Aerosol printing device. (**c**) Preparation process of PDMS/MWCNTs thin films.

**Figure 2 nanomaterials-12-03417-f002:**
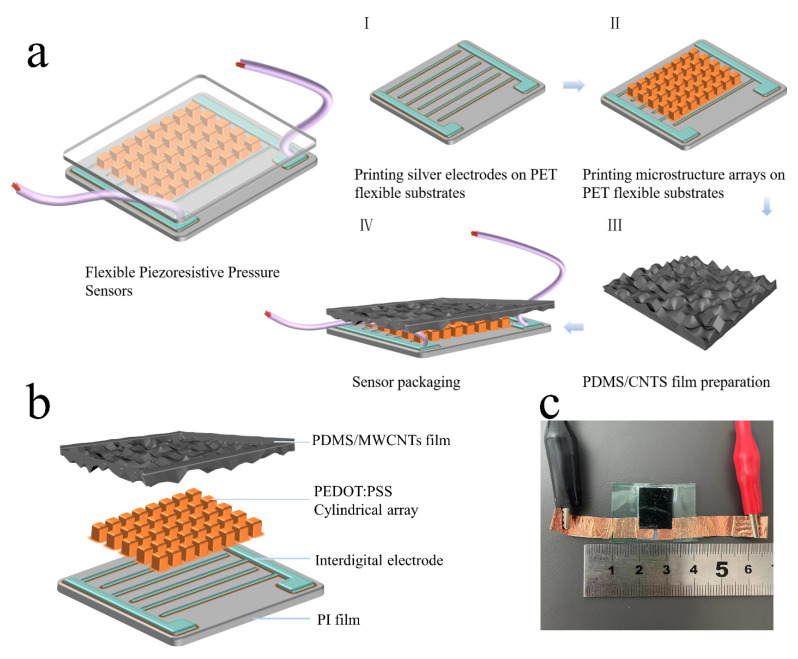
Package and test diagram of the sensor. (**a**) Fabrication process of the sensor. (**b**) Schematic diagram of the sensor structure. (**c**) Physical image of the sensor sample.

**Figure 3 nanomaterials-12-03417-f003:**
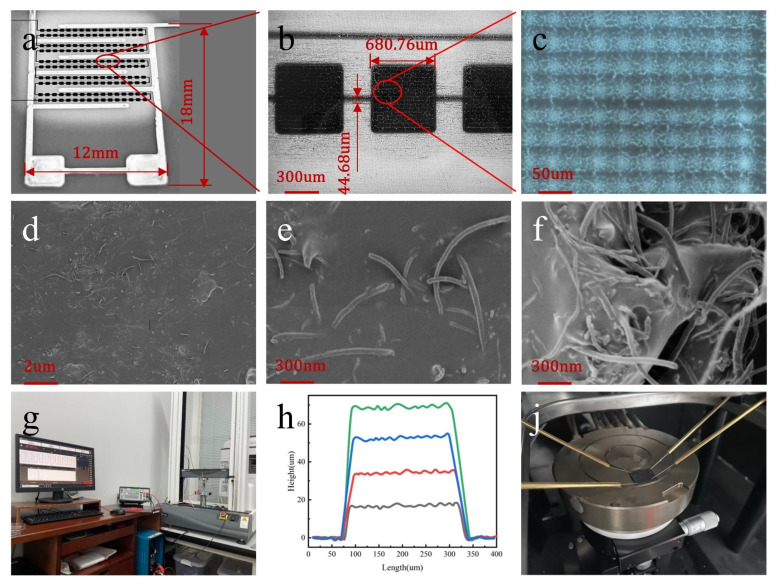
(**a**) Physical image of silver IDE and microstructures printed on PET flexible substrates. (**b**) Microstructure metallographic microscope characterization image. (**c**) Enlarged view of microstructural characterization. (**d**) SEM image of PDMS/MWCNTs thin film. (**e**) Enlarged image of PDMS/MWCNTs film characterization. (**f**) Cross-sectional characterization of PDMS/MWCNTs films. (**g**) Keithley6510 and electronic universal material testing machine test chart. (**h**) Microstructure step meter test image. (**j**) Square resistance test of PDMS/MWCNTs films.

**Figure 4 nanomaterials-12-03417-f004:**
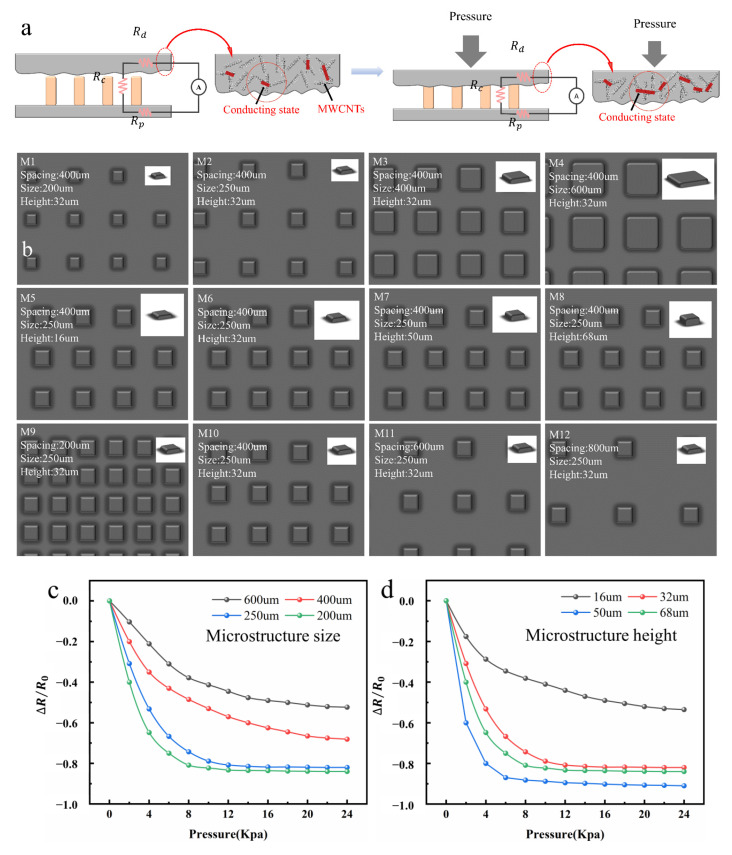
(**a**) Simplified diagram of sensor resistance analysis. (**b**) Dimensions of the M1-M12 sensor dielectric layer microstructure; the text in the figure is the square size, spacing, and height. (**c**) Effect of M1-M4 microstructure size on sensor sensitivity. (**d**) Effect of M5-M8 microstructure height on sensor sensitivity. (**e**) Effect of M9-M12 microstructure spacing on sensor sensitivity. (**f**) M6 multiple-cycle force test. (**g**) M6 cyclic force test response time. (**h**) M6 cycle force test recovery time.

**Figure 5 nanomaterials-12-03417-f005:**
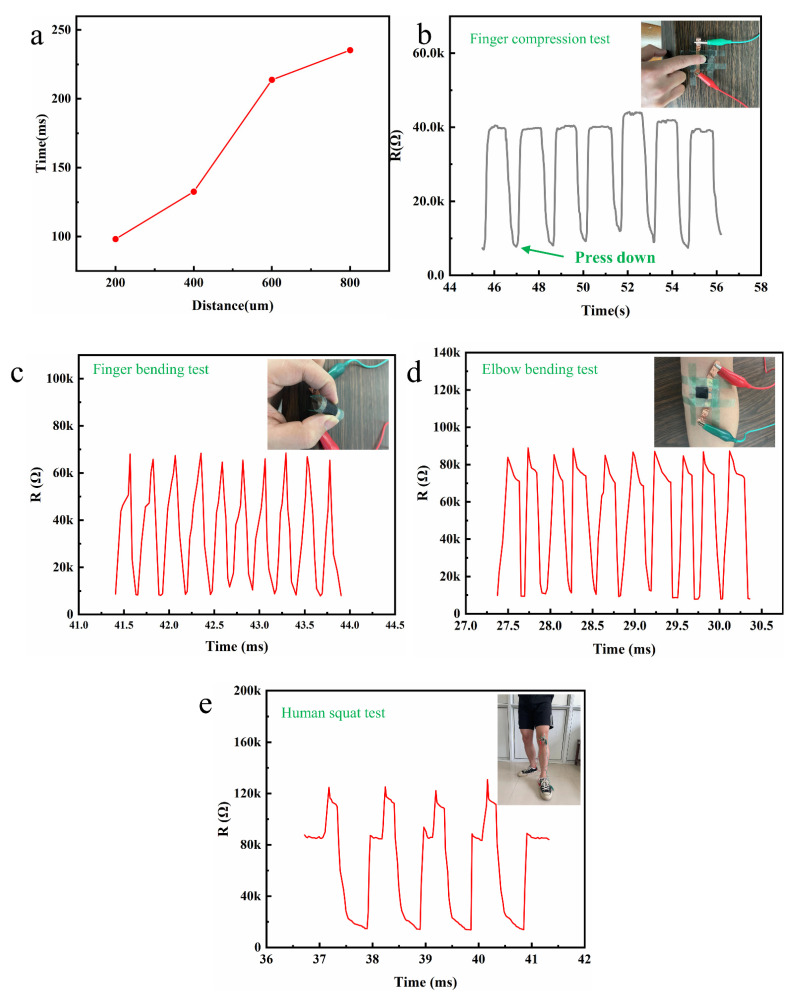
(**a**) Effect of M9-M12 microstructure spacing on relaxation time. (**b**) Intermittent finger compressions. (**c**) Finger bending test. (**d**) Elbow bending test. (**e**) Human squat test.

## Data Availability

The data that support the findings of this study have not been made available but can be obtained from the author upon request.

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
