# Peer review of "High-Performance Flexible Piezoresistive Pressure Sensor Printed with 3D Microstructures"

_nanomaterials, 2022, doi:10.3390/nano12193417_

Round 1
Reviewer 1 Report
In this paper, the authors mentioned as “High Performance Flexible Piezoresistive Pressure Sensor Printed with 3D Micro-structures”. In this manuscript, the authors focused on a new method with aerosol printing technology for preparing high-sensitivity flexible piezoresistive pressure sensors. I think that your focus points on fabrication of high performance of flexible piezoresistive pressure sensor in the field of wearable electronic are important. However, Unfortunately, this paper is not well-formed as an academic paper. Also, it is difficult to understand the data and actual experiments from the manuscript. For example, some figure captions are missing, and explanatory text is missing from the main manuscript. In my opinion, there are unclear points in the main manuscript as the following;
1. In Fig.1, you showed “Aerosol printing device”. However, it is difficult to imagine the actual application area only from the appearance. Please refer to Fig.1a and replace the photos.
2. In Fig. 2c, Fi.3. a, j, there is no scale bar.
3. In Fig.4, you measured response time and recovery time in your device. But I cannot understand why their value were not match?
4. In Fig.4, you showed some data from (a) to (h) about characteristics of your sensor devices. However, I cannot find detail information of Fig. 4(f) to (h) in main manuscript. The authors should add more description about above.
5. In Fig. 5(b) to (e), inset photos are very hard to see because of very small size.
6. In Fig.5 (b) to (e), the authors provided response data when various pressures were applied to your sensor. However, I (and maybe readers) cannot understand how to view each data because there is only response data. Please explanation into the figures. Also, the captions are only (a) and (b), so I can't understand them.
Author Response
Dear reviewer,
The response you can find in the attachment is provided point-by point. If you have any problems, please do not hesitate to contact me. Thank you for your positive comments and valuable suggestions.
Best wishes,
Dr./Prof. Huang

Reviewer 2 Report
The authors present an interesting application of printing technology for creating a fairly novel pressure sensor. Prior to publication, the authors should address the following:
- The text could use some significant proofreading and editing throughout. The second sentence of 2.1, for instance, is long and difficult to follow, and it seems to be missing some words.
- Define EHD.
- Section 2.2.2 states that the Young's modulus of PDMS was 2GPa. Is this a typo? 2MPa seems more reasonable.
- The text section describes how PDMS/CNT was spin coated on sand paper, but it is not well represented in Figure 1c.
- Figure 2 can be a bit confusing. On step III, the PDMS/CNT thin film is represented as a yellow textured surface, but it is not obvious where it fits in step IV. In fact, Figure 4 may be the first spot where the sensor and its working principle is shown. The reader would benefit from something like figure 4a earlier in the text to give the fabrication section more context, perhaps including labels describing which parts are what material (PET, silver ink, PEDOT:PSS, PDMS/CNT).
- Figure 4h should be more explicitly explained.
- Typo in equation 1 where it should be DeltaR/R0 instead of DeltaP/R0.
- Figure 4c, d, e would benefit with a label or cartoon indicating what variable is being changed.
- The authors should note that, although some designs result in higher initial sensitivities, the sensors "bottom out" more rapidly (smaller sensor pressure range) when the sensitivity is high. For example, in Figure 4c, the 600um sample has a higher sensitivity than the 200um sample after about 8KPa of pressure.
- In general, the discussion section is difficult to follow. In some places, this is because of language choices such as referring to the microstructures as cylinders with diameters when they are more rectangular in shape. In most places, the text would greatly benefit from more explicitly explaining why the sensors behave the way they do. For example:
- Why did the sensitivity decrease for samples M5-M7 when height increased to 68um?
- An "air gap rate" and/or "air porosity" are stated to have a "certain impact on the sensor sensitivity." Elaborate on this and how it is related to sensitivity.
- How many samples were tested? Are these results repeatable?
- Figure 5 is missing the captions for c, d, and e. Also, the inset images are so tiny that they are nearly useless to the reader.
- The authors could do more to address the tendency for conductive rubbers to exhibit interesting behaviors due to all sorts of stimuli. Is your sensor sensitive to temperature or humidity? Do you observe any hysteresis? Does the resistance vary when testing at a higher number of cyclic loads?
Author Response

(The authors gave the same response as above.)

Round 2
Reviewer 1 Report
The authors sincerely responded to the comments. The modified manuscript can be accepted to the journal. However, there are still concerns. For example, In Fig.5b the data overlapped with the word. I strongly recommend that you carefully correct such points until publication.
Author Response
Dear editor,
Thank you for your professional suggestion, and we have carefully revised the figures in the manuscript. If there are any other modifications we should make, we would like very much to modify them and we really appreciate your help. Thank you very much.
Dr. Yu
